

# Fast control of interactions in an ultracold two atom system: Managing correlations and irreversibility

Thomás Fogarty[1*], Lewis Ruks[1], Jing Li [1] and Thomas Busch[1]

**1** Quantum Systems Unit, Okinawa Institute of Science and
Technology Graduate University, Onna, Okinawa 904-0495, Japan

⋆ thomas.fogarty@oist.jp

## Abstract

We design and explore a shortcut to adiabaticity (STA) for changing the interaction strength between two ultracold, harmonically trapped bosons. Starting from initially uncorrelated, non-interacting particles, we assume a time-dependent tuning of the inter-particle interaction through a Feshbach resonance, such that the two particles are strongly interacting at the end of the driving. The efficiency of the STA is then quantified by examining the thermodynamic properties of the system, such as the irreversible work, which is related to the out-of-equilibrium excitations in the system. We also quantify the entanglement of the two-particle state through the von Neumann entropy and show that the entanglement produced in the STA process matches that of the desired target state. Given the fundamental nature of the two-atom problem in ultracold atomic physics, the presented shortcut can be expected to have significant impact on many processes that rely on inter-particle interactions.

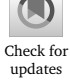
# 1   Introduction

With the increasing complexity of quantum systems, from trapped cold atoms and ions to superconducting quantum circuits, precise and fast control has become of paramount importance. This has led to the development of techniques in optimal control [1], machine learning [2] and shortcuts to adiabaticity (STA) [3], which aim to minimize losses, and to reduce noise and unwanted excitations from dynamical operations on quantum states. For the latter, non-interacting and mean-field systems have been fertile areas of exploration [4–10], while recent forays into interacting many-body systems has shown interesting developments in theory [11–21] and experiments [22, 23], as have applications in entanglement creation and maximization [24–30].

Indeed, controllable contact interactions between neutral atoms in microtraps have been identified as a suitable process for creating conditional entanglement between two atoms [31–33]. Therefore, in this work we focus on controlling the interaction to deterministically create continuous-variable entanglement between two harmonically trapped interacting atoms. This paradigmatic model possesses exact solutions in three, two and one dimensions [34], and is used to benchmark the effect of interactions in few-body systems [35–37] due to the ability to tune interactions effectively with Feshbach resonances [38]. We consider the one-dimensional case and propose creating entanglement between these two particles through the design of a time-dependent Feshbach pulse [39–42]. While the hasty implementation of such a process can have unwanted consequences in relation to the creation of unwanted dynamical processes and fluctuating entanglement [43, 44], we show that a suitably designed STA process can negate these effects even when it is carried out abruptly on short timescales. To design the form of the Feshbach pulse such that it fulfills the desired dynamical evolution of the interacting state we use an inverse engineering method by ultilizing a variational ansatz. Since this ansatz is by its nature only an approximation to the exact state of the system during the evolution, we characterise its effectiveness through calculations of the associated thermodynamical and correlated properties of the system. We also compare the STA to a simple reference drive that has not been optimized to suppress excitations and use this to gauge our results.

The manuscript is organised as follows: in Section 2 we outline the exact solutions to the interacting two particle model and in Section 3 we describe the STA and how its effectiveness can be quantised through the irreversible work. In Section 4 we discuss how this interaction ramp can create entanglement between the two particles and how to characterise irreversible dynamics in this setting. Finally, we conclude.

# 2   Model

The model we consider describes a one-dimensional system of two interacting bosons of mass $m$ confined in an external harmonic trapping potential of frequency $\omega$, for which the Hamiltonian can be written as

$$H = \sum_{j=1}^{2} \left( -\frac{\hbar^2}{2m}\nabla_j^2 + \frac{1}{2}m\omega^2 x_j^2 \right) + V_{\text{int}}(x_1, x_2) \,, \tag{1}$$

where $V_{\text{int}}(x_1, x_2)$ describes the form of the interaction between the particles. At low temperatures the interactions are short-ranged and can be described with a delta-function pseudo potential $V_{\text{int}}(x_1, x_2) = g_{1D}\delta(x_1 - x_2)$, where the interaction strength is characterized by the atomic s-wave scattering length, $a_{3D}$, via $g_{1D} = \frac{4\hbar^2 a_{3D}}{md_\perp^2}\frac{1}{1 - C\frac{a_{3D}}{d_\perp}}$. Here $\omega_\perp$ is the trap frequency in the transverse directions of a quasi-one dimensional trap and $d_\perp = \sqrt{\hbar/m\omega_\perp}$ is the associated

width. The constant $C$ is given by $C = \zeta(\frac{1}{2}) \approx 1.4603$ [45]. In the following, we scale all lengths by $a = \sqrt{\hbar/(m\omega)}$, all energies by $\hbar\omega$, the interaction as $g = g_{1D}/(\sqrt{2}a\hbar\omega)$ and give the time $t$ in units of the inverse trapping frequency, $\omega^{-1}$.

The Hamiltonian for this two particle problem can be solved by moving to centre of mass, $X = \frac{x_1+x_2}{\sqrt{2}}$, and relative coordinates, $\tilde{x} = \frac{x_1-x_2}{\sqrt{2}}$, which uncouples the dynamics and leads to two independent Hamiltonians

$$H(X) = -\frac{1}{2}\frac{\partial^2}{\partial X^2} + \frac{1}{2}X^2 \,, \tag{2}$$

$$H(\tilde{x}) = -\frac{1}{2}\frac{\partial^2}{\partial \tilde{x}^2} + \frac{1}{2}\tilde{x}^2 + g\delta(\tilde{x}) \,. \tag{3}$$

The Hamiltonian for the centre of mass coordinate describes a single particle in a harmonic trap and its solution is readily given by $\psi_n(X) = N_n \mathcal{H}_n(X)e^{-X^2/2}$ with energies $E_n = (n+1/2)$, where $\mathcal{H}_n(X)$ is the $n$-th order Hermite polynomial and $N_n$ is a normalisation constant. Meanwhile, the Hamiltonian for the relative coordinate describes a single particle in a harmonic trap which is split centrally by a delta-function potential. For the odd states of the relative Hamiltonian, $(n = 1, 3, 5, \dots)$, the delta-function potential plays no role due to the eigenstates possessing a node at the trap centre and the solutions are identical to the ones of the quantum harmonic oscillator given above. For the even states, $(n = 0, 2, 4, \dots)$, the solutions are more complex and are given by $\phi_n(\tilde{x}) = N_n e^{-\tilde{x}^2/2}U(1/4 - E_n/2, 1/2, \tilde{x}^2)$, where the $U(a, b, z)$ are the Kummer functions [34]. The energy of the even states can be found by solving the implicit equation

$$-g = 2\frac{\Gamma\left(-\frac{E_n}{2} + \frac{3}{4}\right)}{\Gamma\left(-\frac{E_n}{2} + \frac{1}{4}\right)} \,. \tag{4}$$

In the following we will choose our initial state to be the groundstate of the two-particle system, $\Psi(x_1, x_2) = \psi_0(X)\phi_0(\tilde{x})$, with all possible other initial states being straightforward extensions. We are interested in creating an interacting two particle state over a short time interval and with no excitations at the end of the Feshbach pulse. This means that our initial state is a separable state with $g_i = 0$ at $t = 0$, while at the end of the interaction ramp, at $t = t_f$, the interaction between the two particles is at a fixed, pre-determined value $g_f$. This process is shown in Fig.1 with the initial uncorrelated two-body state being Gaussian in nature (panel(a)), and after application of the chosen interaction ramp $g(t)$ (panel (b)) the strongly interacting final state is achieved (panel(c)). Here the strong interaction ($g_f = 20$) between the particles significantly reduces the probability density between the particles along $x_1 = x_2$, forming two distinct lobes in position space. Since our system can at any point be separated into centre-of-mass and relative components, the effects of the interaction ramp $g(t)$ only apply to the relative dynamics and we can ignore the dynamics of the centre-of-mass part of the wavefunction.

## 3   Shortcut to adiabaticity

The exact functional time-dependence of the chosen $g(t)$ will have consequences for how close the dynamical state, $\Psi(x_1, x_2, t_f)$, is to the target equilibrium state, $\Psi_T(x_1, x_2)$, at the end of the interaction ramp. To minimize unwanted excitations, we design an STA using the method of inverse engineering whereby the effective Lagrangian [46] of the system

$$\mathcal{L} = \int_{-\infty}^{\infty}\left[\frac{i}{2}\left(\frac{\partial \Phi_c}{\partial t}\Phi_c^* - \frac{\partial \Phi_c^*}{\partial t}\Phi_c\right) - \frac{1}{2}\left|\frac{\partial \Phi_c}{\partial \tilde{x}}\right|^2 - g(t)\delta(\tilde{x})|\Phi_c|^2 - \frac{1}{2}\tilde{x}^2|\Phi_c|^2\right]d\tilde{x} \tag{5}$$

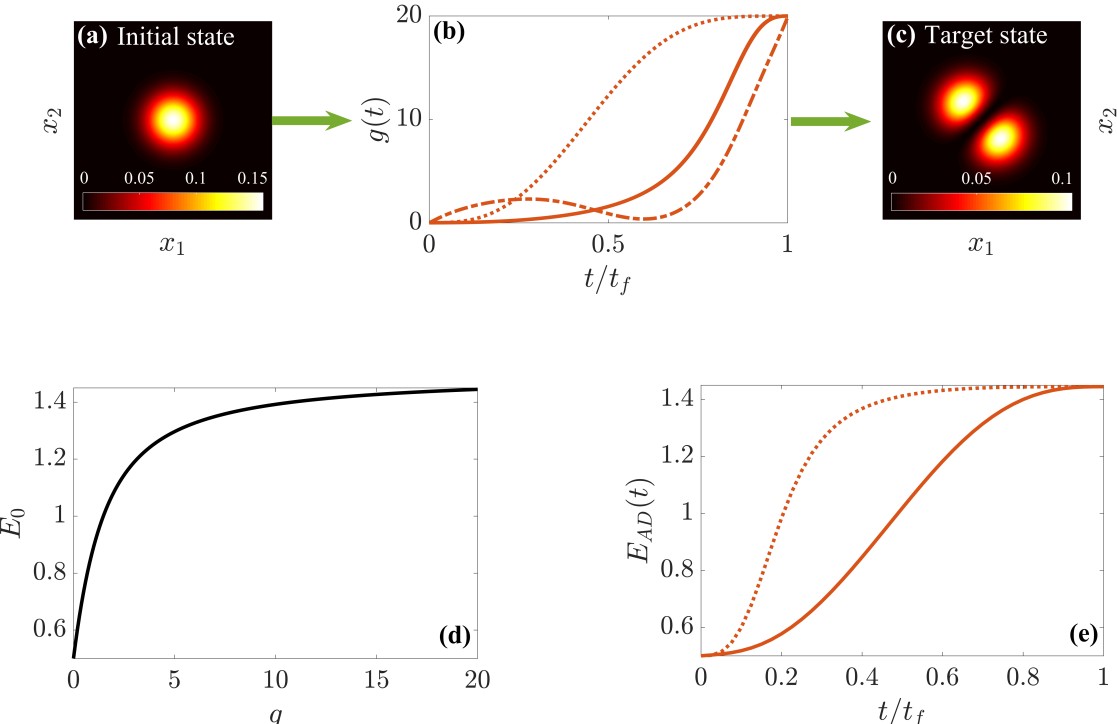

Figure 1: (a) Density $|\Psi(x_1, x_2)|^2$ of the initial state at $g_i = 0$, which is the uncorrelated non-interacting two-particle groundstate. (b) Time dependence of the interaction parameter as given by the STA for $t_f = 10$ (solid line) and $t_f = 1$ (dash-dotted line), and by the reference function given in Eq. (12) (dotted line), for a final interaction of $g_f = 20$. (c) Density of the desired final state at $g_f = 20$. (d) Groundstate energy $E_0$ as a function of the interaction $g$ as calculated from Eq.(4). (e) Adiabatic energy $E_{AD}(t)$ versus time for $g_{\text{STA}}(t)$ (solid line) and $g_{\text{ref}}(t)$ (dotted line).

is minimized with respect to a chosen ansatz for the relative part of the wavefunction. A good choice of ansatz is given by a superposition of the initial and the desired final state

$$\Phi_c(\tilde{x}, t) = \varphi(\tilde{x}, t)e^{ib(t)x^2} = \mathcal{N}(t)\big[(1 - \eta(t))\phi_0(\tilde{x}, g_i) + \eta(t)\phi_0(\tilde{x}, g_f)\big]e^{ib(t)x^2}, \quad (6)$$

where $\mathcal{N}(t)$ is a time dependent normalization and $b(t)$ is a chirp that allows the wavefunction to change its width during the ramping process [46,47]. The switching function, $\eta(t)$, allows to smoothly change from the initial state, $\phi_0(\tilde{x}, 0)$ at $g_i$, to the final state, $\phi_0(\tilde{x}, t_f)$ at $g_f$, by imposing the boundary conditions $\eta(0) = 0$ and $\eta(t_f) = 1$. All parameters are assumed to be real functions of time and we choose $\eta$ to be described by the polynomial $\eta(t) = \sum_{j=0}^{5} a_j t^j$, so that the boundary conditions of the first and second derivatives $(\dot{\eta}(0) = \dot{\eta}(t_f) = \ddot{\eta}(0) = \ddot{\eta}(t_f) = 0)$ can be fulfilled. This ensures that the switching function is smooth and that the phase at the beginning and end of the shortcut process is zero [48].

We minimize the effective Langrangian in Eq. (5) with respect to the chirp $b(t)$ and the width $\xi(t) = \sqrt{\langle \tilde{x}^2 \rangle - \langle \tilde{x} \rangle^2}$ to give

$$\dot{\xi} = 2b\xi, \quad (7)$$

$$\dot{b} = -\frac{1}{2} - 2b^2 - \frac{g_{\text{STA}}(t)\frac{\partial}{\partial \eta}|\varphi(0,t)|^2 + \frac{1}{2}\frac{\partial \beta}{\partial \eta}}{\frac{\partial \xi^2}{\partial \eta}}, \quad (8)$$

where we have defined $\beta = \int_{-\infty}^{+\infty} \left|\frac{\partial}{\partial \tilde{x}}\varphi(\tilde{x}, t)\right|^2 d\tilde{x}$. Note that we assume that there is no trans-

port of particles in the relative Hamiltonian, so that $\langle \tilde{x} \rangle = 0$ and $\langle \tilde{x}^2 \rangle = \int_{-\infty}^{+\infty} \tilde{x}^2 |\varphi(\tilde{x}, t)|^2 d\tilde{x}$.

The interaction strength $g_{\text{STA}}(t)$ can now be derived from Eq. (8) as

$$g_{\text{STA}}(t) = -\frac{\frac{\partial \xi^2}{\partial \eta}\left(\frac{\partial b}{\partial t} + 2b^2 + \frac{1}{2}\right) + \frac{1}{2}\frac{\partial \beta}{\partial \eta}}{\frac{\partial}{\partial \eta}|\varphi(0, t)|^2}. \tag{9}$$

For the integrals that contain hypergeometric functions with arbitrary $g$, no compact exact solutions are known to our knowledge and we therefore evaluate them numerically, however for the case of $g_i = 0$ and $g_f = \infty$ tractable solutions can be found. This situation corresponds to ramping the interaction to the well known Tonks-Girardeau (TG) limit [49] whereby the infinitely strong repulsive interaction causes the relative groundstate wavefunction to become doubly degenerate with the first odd excited state $\phi_1(\tilde{x})$. This allows one to write the relative wavefunction as $\phi_0(\tilde{x}, g_f) = \phi_1(|\tilde{x}|) = (\pi/4)^{-1/4} e^{-|\tilde{x}|^2/2}|\tilde{x}|$, where $|\tilde{x}|$ preserves the even parity, leading to the solutions

$$\mathcal{N}(t) = \sqrt{\frac{1}{1 - 2\eta(1 - \eta)(1 + \sqrt{2/\pi})}}, \tag{10}$$

$$\xi(t) = \mathcal{N}(t)\sqrt{1/2 + \eta - 2\eta(1 - \eta)(\sqrt{2/\pi} + 1)}. \tag{11}$$

From this one can easily obtain $g_{\text{STA}}$ by carrying out the partial derivatives in Eq. (9).

To test the effectiveness of this shortcut, we compare with a reference function described by a generic sinusoidal ramp of the interaction, which is designed to satisfy the boundary conditions $g_{\text{ref}}(0) = 0$, $g_{\text{ref}}(t_f) = g_f$ and $g'_{\text{ref}}(0) = g'_{\text{ref}}(t_f) = g''_{\text{ref}}(0) = g''_{\text{ref}}(t_f) = 0$, as

$$g_{\text{ref}}(t) = \frac{g_f}{32}\left[30\sin\left(\frac{\pi}{2}\frac{t}{t_f}\right) - 5\sin\left(\frac{3\pi}{2}\frac{t}{t_f}\right) - 3\sin\left(\frac{5\pi}{2}\frac{t}{t_f}\right)\right]. \tag{12}$$

We stress that the choice of our reference ramp is to some degree arbitrary, and any smoothly changing function will serve as a suitable comparison to the STA ramp. In this case the form given in Eq. (12) outperforms a simple linear ramp of the interaction on the timescales we will consider.

A comparison between the interaction ramp designed by the STA and the reference function for $g_i = 0$ and $g_f = 20$ with a duration of $t_f = 1$ and $t_f = 10$ is shown in Fig. 1(b). For longer ramp times ($t_f = 10$, solid line) one can immediately see the STA pulse starts out slower and increases faster at the end. While the general form of this solution is mostly a consequence of the form of the ansatz for $\eta(t)$, the exact shape is due to the dependence of the ground state energy on the interaction strength, which is plotted in Fig. 1(d). One can see that this energy increases quickly with increasing values for $g$, and converges to $E_0 = 1.5$ for larger values. Assuming that the energy of the system adiabatically follows a given $g(t)$, we show in Fig. 1(e) how the adiabatic energy $E_{AD}(t)$ changes with the interaction pulses for the STA (solid line) and the reference ramp (dotted line). Increasing $g$ slowly initially, as suggested by the STA pulse, therefore ensures that the energy grows at a slow rate, which can be maintained later by increasing $g$ faster. In contrast, the non-optimised reference ramp (dashed line) causes a sudden and large increase in energy at the beginning of the interaction pulse and therefore could lead to more irreversibility. The rate of change of energy, and not of the interaction strength, is therefore the quantity which will determine the success of the individual interaction pulses in producing the desired final state. For shorter interaction ramps ($t_f = 1$, dash-dotted line) the STA pulse develops distinct modulations and for ramp durations $t_f < 1$ the interaction strength can even become negative during certain parts of the STA

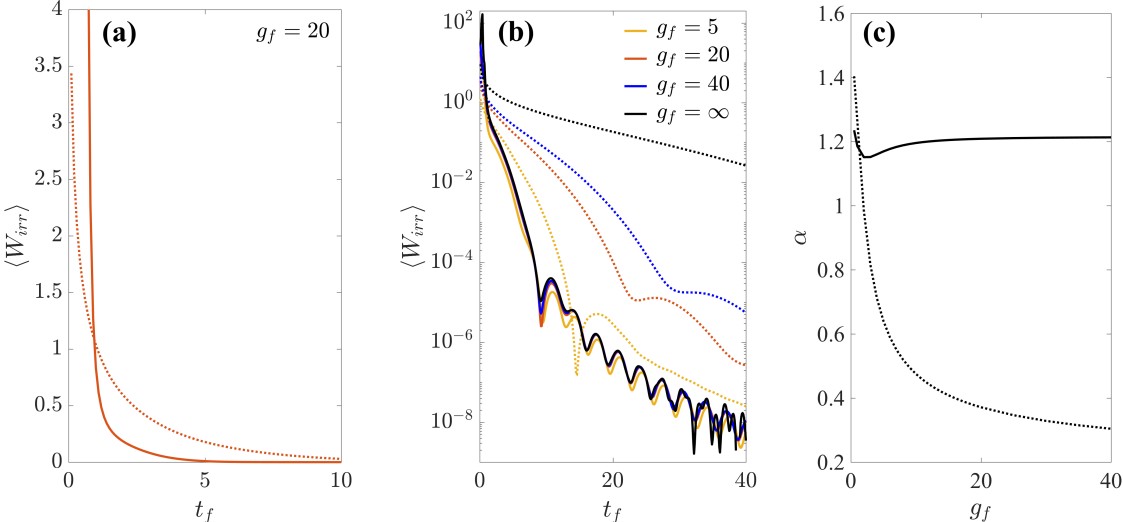

Figure 2: (a) Irreversible work as a function of the ramp time $t_f$ for a final interaction of $g_f = 20$, using the STA (solid line) and reference ramp (dotted line). (b) $\langle W_{\mathrm{irr}} \rangle$ in log scale versus $t_f$ for $g_f = 5$ (yellow), $g_f = 20$ (red) and $g_f = 40$ (blue). The TG limit, $g_f = \infty$, is given by the black lines, and is exact for the STA pulse, but has been approximated for the reference pulse by choosing $g_f = 1000$. (c) The decay rate $\alpha$ as a function of final interaction $g_f$. In all panels the solid lines represents the STA, while the dotted lines represent the reference ramp.

pulse. This has consequences for the performance of the STA as the system is driven through the resonance point [12, 13].

The standard way to quantify the performance of an STA is to calculate the overlap $|\langle \phi(\tilde{x}, t_f)|\phi_T(\tilde{x})\rangle|^2$ between the state at the end of the dynamics, $\phi(\tilde{x}, t_f)$ , and the target state, $\phi_T(\tilde{x})$. However, in this case the fidelity can be misleading, as the overlap between the dynamical state and the target state is always finite and large due to the wavefunction only being affected locally by the delta-function potential. For this reason we suggest to assess the effectiveness of the STA by calculating the irreversible work at the end of the interaction ramp ($t = t_f$) which is given by

$$\langle W_{\mathrm{irr}} \rangle = E(t_f) - E_T, \tag{13}$$

where $E(t_f)$ is the energy of the system after the ramp, while $E_T$ is the energy of the target equilibrium state. For adiabatic processes $\langle W_{\mathrm{irr}} \rangle$ will vanish, while for non quasi-static processes it quantifies the excess energy in the system as a result of the non-equilibrium excitations created during the time-dependent protocol. Successful implementation of the STA will ensure that all these unwanted excitations are suppressed and the target adiabatic state is reached.

We show the irreversible work as a function of $t_f$ for the STA ramp (solid line) and the reference function (dotted line) for $g_f = 20$ in Fig 2(a). One can see that the use of the STA out-performs the reference ramp at intermediate and long times and reaches the adiabatic limit at about $t_f \approx 5$. However, for very short times ($t_f \lesssim 1$) the amount of $\langle W_{\mathrm{irr}} \rangle$ produced by the STA diverges rapidly, as the system is driven far from equilibrium. In fact, on this timescale the interaction strength becomes negative, $g_{\mathrm{STA}}(t) < 0$, and in this situation our ansatz is no longer suitable, as it explicitly assumes that the interaction between the particles is always repulsive [12]. This places a limit on the range of validity of our approach to ramp times $t_f > 1$ irrespective of the value of $g_f$. At such short times the reference ramp is to a good approximation a sudden quench and therefore does not possess the same rapidly growing

modulations as the STA, thereby producing less irreversible work [13].

The amount of irreversible work produced during the process for different final interactions strengths is shown in Fig. 2(b) for $g_f = \{5, 20, 40, \infty\}$. It is immediately apparent that for the reference ramp the amount of $\langle W_{\text{irr}} \rangle$ produced increases significantly for increasing $g_f$, as driving to stronger interactions in a short time heightens the probability of exciting the system to higher energy modes. In fact, sudden interaction quenches to the TG limit of infinite interactions have been shown to possess a diverging energy expectation value [50, 51], and while the finite time quenches considered here have finite expectation values, there also exists significant excess energy when ramped to the TG limit. In comparison, the STA ramps work effectively irregardless of the interaction strength, even when driven to the TG regime, and it possesses approximately equivalent dynamics over a wide range of $t_f$ (all the solid lines overlap). This is not surprising as the ansatz we have chosen in Eq. (6) and the form of $\eta(t)$ do not implicitly depend on the temporal evolution of the interaction strength. In fact, the system goes through a succession of states that try to keep the amount of irreversible energy low, and therefore mimic an adiabatic evolution closely.

For ramp times longer than $t_f > 10$ finite size effects appear as oscillations in $\langle W_{\text{irr}} \rangle$, while for short ramp times, $t_f < 10$, we find that the irreversible work decays effectively exponentially as $\sim e^{-\alpha t_f}$. We numerically extract these decay rates $\alpha$ from the irreversible work and in Fig. 2(c) show their dependence on the final interaction $g_f$. The STA exhibits the expected stability across the entire range of interactions, while the reference ramp possesses a strong dependency on the magnitude of the final interaction strength with the decay rate reducing significantly for large values. This is consistent with the observations in Fig. 2(b).

# 4 Efficient creation of entanglement

In the previous section we have used the irreversible work to quantify the efficiency of the suggested STA and have shown that nonequilibrium excitations can be successfully suppressed during fast interaction ramps. Even though quantum thermodynamical properties can in certain circumstances be linked to the existence of non-classical correlations [43, 52–54], it is not straightforward to conclude that the STA produces the same continuous variable entanglement as inherent in the desired final state [40, 41, 55]. To calculate the entanglement of our two-particle pure state we use the von Neumann entropy (vNE) which is a well defined measure of entanglement, being zero if and only if the state is a product state [56, 57]. It is given by

$$S = -\sum_n \lambda_n \log_2 \lambda_n. \tag{14}$$

Here the $\lambda_n$ are the eigenvalues of the reduced single particle density matrix (RSPDM)

$$\rho^1(x, x') = \int_{-\infty}^{\infty} \Psi(x, x_2)\Psi(x', x_2)dx_2, \tag{15}$$

which describes the self-correlation of a single particle after tracing out all other particles. The number of non-zero eigenvalues is the Schmidt rank and if it is larger than 1, the two-particle state is entangled. The vNE can also be used as a measure of irreversibility in the system, similar to the irreversible work, and will therefore be affected by the quantum fluctuations induced by the non-adiabatic interaction ramp [43]. To quantify the ability of the STA to create correlations between the two particles we calculate the difference between the vNE at the end of the interaction ramp, $S(t_f)$, and the vNE of the target equilibrium state $S_T$

$$\Delta S = S(t_f) - S_T. \tag{16}$$

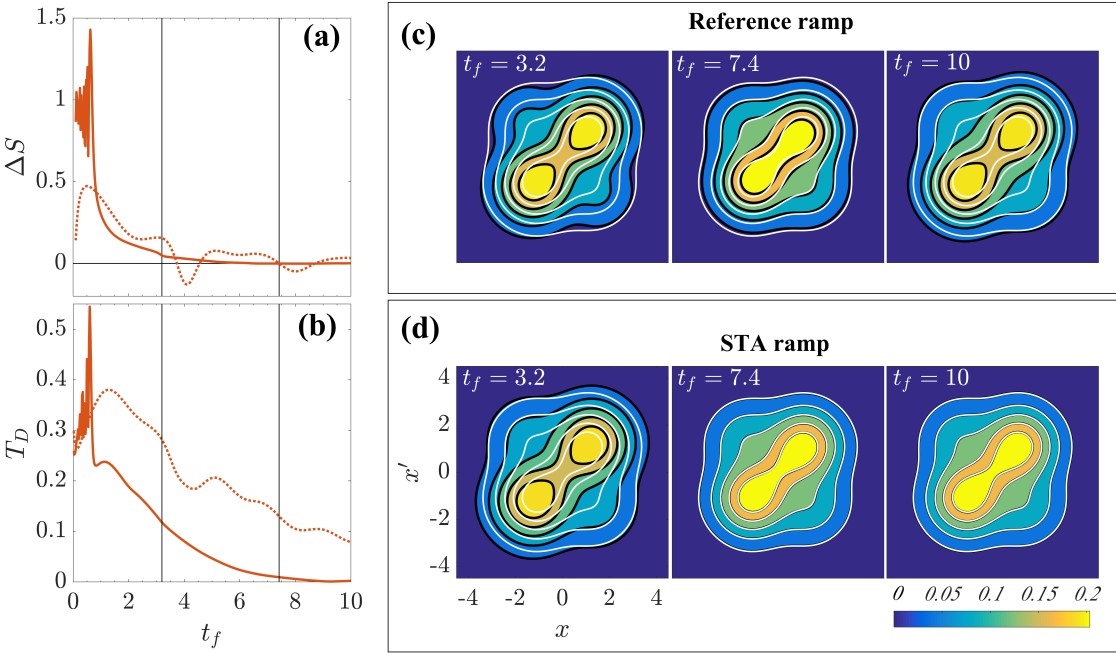

Figure 3: (a) Von Neumann entropy difference $\Delta S$ as a function of ramp times $t_f$, for $g_f = 20$, and (b) the trace distance $T_D$ between the final RSPDM after the ramp and the target RSPDM. The results from the STA are shown as solid lines, while the ones from the reference pulse are shown as dotted lines. (c-d) RSPDM at different ramp times $t_f$ for the reference ramp and the STA ramp. The target RSPDM is overlaid on each plot as the white contour lines and the colour scale is the same in each figure.

Since for adiabatic ramp one would expect that $\Delta S = 0$, while the irreversible dynamics should result in $\Delta S \neq 0$, this quantity is an analog irreversible vNE to the already discussed irreversible work.

The entanglement difference as a function of $t_f$ for $g_f = 20$ is shown in Fig. 3(a). One can immediately notice the presence of distinct oscillations at longer times for the reference ramp, which are not present in the irreversible work. These can be attributed to the breathing dynamics of the two particles in the harmonic trap, where the particles periodically collide at the trap centre [43, 58–60]. In this process the separation between them is changed, which has a sizeable effect on the vNE and consequently on $\Delta S$. In comparison, implementing the STA suppresses the breathing dynamics and this irreversible entropy decays exponentially at a rate similar to $\langle W_{\text{irr}}\rangle$.

While the STA successfully creates the target state for times $t_f > 5$, one can notice special ramp times for which the reference pulse also possesses no irreversible vNE ($\Delta S = 0$ at $t_f = \{3.75, 4.55, 7.4, 8.75\}$). To examine if the state produced after these interaction ramps is truly equivalent to the target state, we examine the RSPDM at three different ramp times $t_f = \{3.2, 7.4, 10\}$ (indicated by the vertical lines in panel (a)). In panel (c) the RSDPM is shown after the reference ramp, in panel (d) the corresponding RSPDM after the STA, and for comparison the RSPDM of the target state is shown superimposed in each panel (white contours). At $t_f = 3.2$ we have $\Delta S > 0$ for both the reference and STA, and therefore the respective RSPDMs are quite different from the target RSPDM. At $t_f = 10$ the STA fully recreates the adiabatic dynamics with $\Delta S = 0$ and has a RSPDM identical to the one of the target state, while the reference ramp has $\Delta S > 0$ and quite a different RSPDM. At the special ramp time of $t_f = 7.4$, for which the STA pulse and the reference pulse show $\Delta S \approx 0$, indicating that no irreversible vNE is created in either protocol, the RSPDM of the STA is identical to the target

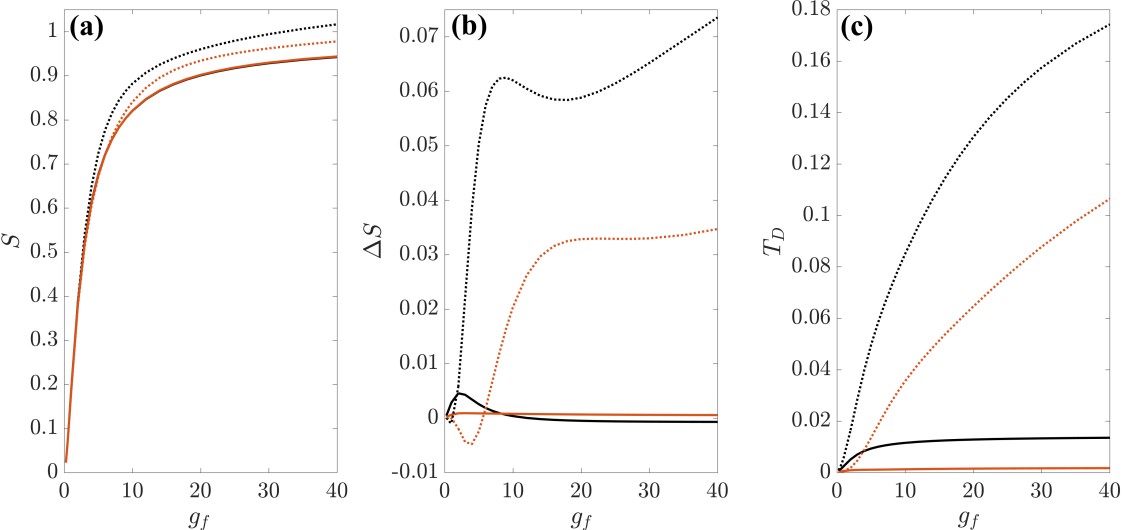

Figure 4: (a) Von Neumann entropy as a function of the final interaction $g_f$ for $t_f = 6.6$ (black lines) and $t_f = 10$ (red lines), for the reference protocol (dotted lines) and STA (solid lines). The entropy after the STA for both of the ramps times are on top of each other in this figure. (b) The von Neumann entropy difference $\Delta S$ and (c) the trace distance $T_D$ between the RSPDM after the ramp and the target RSPDM, for the same parameters as panel (a).

RSPDM while the RSPDM of the reference shows a discernible difference. This indicates that some irreversibility is still present in the reduced state which is not captured by the vNE alone.

We therefore use the trace distance $T_D(t_f) = \frac{1}{2}\text{Tr}\left[\sqrt{(\rho(t_f) - \rho_T)^2}\right]$ to quantify the difference between the reduced density matrix after using the STA or reference protocol and that of the target reduced state $\rho_T$. If $T_D = 1$ the two density matrices are orthogonal, while for $T_D = 0$ the two density matrices are identical. In Fig. 3(b) the trace distance is shown as a function of $t_f$ for the STA and the reference protocol and, contrary to the vNE difference, it is always finite when using the reference ramp. This means that the RSPDM is far from that of the target state for all examined timescales, and the zeros observed in panel (a) do not indicate a successful creation of the target state, but rather a different state with the same amount of entanglement. One can also see that the trace distance for the STA vanishes only at $t_f \approx 10$, which is later than indicated by the vanishing of the difference in the vNE.

Lastly, we study the behaviour of the entanglement and the trace distance as a function of final interaction strength. The von Neumann entropy is shown for two different ramp times ($t_f = 6.6$ (black) and $t_f = 10$ (red)) in Fig. 4(a), where the STA protocol (solid lines) is compared to the reference one (dotted lines). The effect of the contact interaction on the entanglement between two bosons has been well studied [40–42] and it is known that the entanglement increases linearly for low interactions, while for large values of $g$ it saturates and asymptotically approaches that of a TG molecule ($S = 0.985$). Using the STA we succinctly capture this behaviour and find that for both values of $t_f$ shown here the entanglement almost exactly matches that of the corresponding equilibrium state for any $g$. In comparison, the reference protocol does not follow the same behaviour, it only creates comparable values of the vNE for small $g$, while at stronger interactions it diverges from the equilibrium state due to the presence of the non-adiabatic excitations discussed in the previous sections.

This growing irreversibility of the reference ramp with larger interaction strengths is also visible in the irreversible vNE ($\Delta S$ in panel (b)), and in the trace distance ($T_D$ in panel (c)), where large deviations from the equilibrium state exist when the system is ramped to large

interactions ($g_f > 10$). Contrary to this the STA ramp achieves consistent results over the whole range of interaction strengths, highlighting the robustness of our approach. Employing the STA shows a significant improvement over the reference ramp, confirming the observations from the irreversible work, and further substantiating that the STA protocol can be used to generate fast and frictionless entanglement between two particles.

## 5 Conclusion

We have designed an STA to efficiently tune the interaction between two ultracold bosons, which can be used to deterministically create non-classical correlations on short timescales. Although our approach is based on a variational ansatz, the effectiveness of the STA has been confirmed by calculating different quantifiers, namely, the irreversible work, the von Neumann entropy and the trace distance, and shows promise for using similar approximate techniques for more complex interacting systems. Specifically, our choice of ansatz shows remarkable consistency over a wide range of interactions and it suitably outperforms a typical reference function. This is the result of our choosing the ansatz to be a time dependent superposition of the initial and the desired target state, which can also be a successful strategy in systems which possess no distinctive time dependent parameters over which to optimise [48].

While we have only shown results for an increase in interactions starting at $g_i = 0$, increasing or decreasing the interaction from an initial finite value of $g$ is also possible. Additionally, generalisations to higher dimensions are straightforward (even in non-isotropic traps), however they might become tedious.

The system we have considered is experimentally realisable in modern ultracold atom experiments and has in the last two decades become the paradigmatic interacting few-body system: two harmonically trapped ultracold atoms. The existence of the two-particle shortcut is therefore likely to lead to the development of shortcuts for larger systems of interacting particles, where similar ansatz can be employed [61–63]. The successful suppression of oscillations in the vNE points to the possibility to create stable entanglement in a highly precise manner, which may be used to realise high fidelity collisional quantum gates [33, 64, 65].

Finally, we have shown that it is important to consider different benchmarks for quantifying the success of the STA in quantum systems beyond the standard fidelity. While all quantities showed that the STA successfully suppresses unwanted excitations compared to the reference function, each captured qualitatively different dynamics according to the chosen quantity. This fact will become even more important when describing larger interacting many-body systems.

## Acknowledgements

The authors acknowledge non-adiabatic discussions with Steve Campbell.

**Funding information** This work was supported by the Okinawa Institute of Science and Technology Graduate University. TF acknowledges support under JSPS KAKENHI-18K13507.

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
