# Peer review of "Fast control of interactions in an ultracold two atom system: Managing correlations and irreversibility"

_SciPost Physics, doi:SciPost Phys. 6, 021 (2019)_

## Round 1 · Referee Report · Anonymous · 2018-7-5

Strengths

I think this is a conceptually important addition to the literature of STA given that its focus on fast nonadiabatic interaction tuning.

This is highly relevant for many applications in quantum simulation via digital and analogue techniques. It is directly applicable to ultracold atoms where interactions can be tuned via Feshbach resonances or CIRs. It is also a nice addition to previous work by the authors on efficient nonlinear engine with a soliton as a working substance as the current description goes beyond mean-field.

Weaknesses

The main limitation of the work is the fact that it is restricted to N=2.

Report

My only recommendation is that the authors consider the potential generalization of the problem treated in this manuscript to N-body problems.

The context for that is clear. Interaction quenches have been exploited, e.g., to prepare metastable states such as the super-TG gas. Though this is contrary to the previous experimental efforts, could one envisioned to prepare the ground state of attractive bosons by this method? More generally, the goal would be to prepare bound states “in the ground state” without exciting other modes (vibrational, breathing, etc.).

A more tractable problem would deal with the preparation of the ground state of a TG gas starting from an ideal Bose gas by ramping the interactions in finite time. It would be easy to generalize the ansatz in Eq. (7). Perhaps the ansatz by Girardeau would be interesting in this context [Phys. Rev. Lett. 91, 040401 – Published 24 July 2003].
Indeed, for N-particle systems it may pay off to consider a different kind of ansatz based on interpolating between k-body-reduced density matrices instead of many-body wavefunctions. This would suffice to match k-body correlations. The case of one-body reduced density matrices (OBRDMs) would already be very interesting and of comparable complexity to the problem discussed by the authors.

On a different note, would it be fair to say that the work also presents a way of controlling entanglement?

Requested changes

After the authors have considered the optional suggestions above, I recommend the publication of the manuscript in Sci. Post.

  • validity: good
  • significance: high
  • originality: good
  • clarity: good
  • formatting: good
  • grammar: excellent

Author:  Thomas Fogarty  on 2018-07-12  [id 291]

(in reply to Report 1 on 2018-07-05)
Category:
answer to question
suggestion for further work

We’d like to thank the referee for reading our manuscript and his/her report. We are happy to see that they think that our work deserves to be published in Sci. Post and are grateful for their suggestions. Below we respond in detail to each point made in the report.

We fully agree with the referee that the extension of the presented STA protocol to larger systems is very interesting. In fact, it is a direction we are currently working on.
As the referee points out, this is firstly a problem of finding an appropriate ansatz and the one mentioned above by Girardeau is certainly interesting. Our first extension, however, is based on the work of Zinner et al. (Sci. Rep. 6, 28362(2016), PRA 95, 053632 (2017)), as it allows to treat few-body systems of bosons and fermions, as well as Bose-Fermi mixtures. As this work is quite involved, and as the two-particle case demonstrates the full breadth of our method, we have decided not to include any details about larger systems in the current manuscript.

Approaching the problem of creating a many-body STA through optimisation of the one-body reduced density matrix (OBRDM) is indeed a very interesting idea. We fully agree that direct manipulation of the OBRDM would allow for more intuitive control of the correlations in the system. Also this approach could potentially allow for easier scaling to larger systems as long as one can calculate the OBRDM efficiently (for instance in the TG regime).

For the final point the referee is correct, indeed we envisaged the work as a means to create stable entanglement between particles, and also to control it via careful application of the Feshbach resonance, which was inspired by proposals of collisional quantum gates [T. Calarco et al, Phys. Rev. A 61, 022304 (2000); O. Mandel et al, Nature 425, 937 (2003), and A. Negretti et al, Quantum Inf Process 10: 721 (2011)]. The successful suppression of oscillations in the von Neumann entropy due to collisions between the particles (as seen in Fig. 3a) points to the possibility to create entanglement in a highly precise manner, which is in contrast to the fluctuating correlations created by the reference pulse. We have amended the revised version of the manuscript to emphasise this point.

---

## Round 1 · Referee Report · Anonymous · 2018-11-5

Strengths

1- The paper deals with a relevant system, ie. the minimal few-body system
2- The paper describes a way to produce correlated two-body states in reasonable times
3- it may be relevant to current experiments
4- Deals with 1D systems, where quantum effects are emphasized

Weaknesses

1- The manuscript presents an approximate solution, its limitations should be clearly stated
2- The information presented does not clearly allow one to asses whether it would be experimentally feasible or not for arbitrary values of t_f
3- The discussion on the entanglement is a bit cumbersome

Report

1) First, it should be emphasized that the current result is not a
theorem, as for instance the shortcut protocol for the 1D harmonic
oscillator by Chen et al. Here the result is somehow euristic, and
has a certain range of validity. This is to me the weakest point
of the current result, I do not see in the manuscript a concise
figure showing when and why does their approach fail. What sets
the minimal time scale t_f which allows their approach to work?

2) In their simulations the authors always start from g=0, does their
approach also work for a finite initial g? A priori I would think that
experimentally it would be easier to initially produce the two
atom system with some interaction, as interaction helps in cooling
down the gas.

3) The authors report one of the STA rampings, in Fig 1b) But then
report many results for different values of t_f. Is the shape of the
ramping unchanged regardless of the value of t_f? Usually, going to
small values of t_f these kind of protocols require fairly involved
time dependences for g(t) which then are almost impossible to
probe experimentally. Is this the case here? In particular, does
their protocol require negative values of g(t) at any time? Does
the value of g(t) become exceedingly large during the protocol?

4) The authors compute the von Neumann entropy from the eigenvalues
of the single particle density matrix. The authors are thus basically
studying whether the system is described by a meanfield or correlations
between the particles are needed. In the beginning of Sect 4 there
seems to be some confusion as the authors talk about bipartite entanglement.
Usually, bipartite entanglement is referrerd to splitting the physical
system in two, say, left and right, and not to what they are reporting, which is more of a degree of condensation, or fragmentation (see the paper by Mueller et al, PRA 74 on Fragmentation of Bose-Einstein condensates). In anycase,
the discussion on the von Neumann entropy and entanglement generation
should be clarified.

5) The authors compare with a reference ramp with a somehow very constrained structure. Probably, a clearer comparison could be made with a linear ramp. Do the authors know whether their reference ramp is better than linear?

Minor issues:

In Fig 3, panels (c) and (d), there is no color bar indicating the
magnitude of what is represented, a scale would be appropriate, what is
the value at which the countour lines a plotted? A similar point can also
be raised with the color plot in Fig 1.

Requested changes

1) A concise figure showing when and why does their approach fail.
2) Explain in detail the features of g(t) for arbitrary protocols to see if it gets too large, negative, etc
3) A figure to show the robutsness of the protocol is needed (what happens if the experiment can only follow their path to within 5%)
4) Clarify the entanglement computed

  • validity: good
  • significance: good
  • originality: high
  • clarity: good
  • formatting: excellent
  • grammar: excellent

Author:  Thomas Fogarty  on 2018-11-20  [id 348]

(in reply to Report 2 on 2018-11-05)
Category:
answer to question

We would like to thank the referee for their careful reading of our manuscript and their helpful suggestions. Below we address each of the comments individually.

1) As our protocol is based on a variational ansatz, the referee is correct in pointing out that it is not a theorem. However, we clearly show that this approximate approach is a good way to obtain a highly-effective STA for a nontrivial system. Indeed, when dealing with non-integrable systems with realistic interactions which, in particular, do not possess scale invariant solutions [see for example Deffner et al, PRX 4, 021013], approximative methods are the only techniques known. This has also been found in other approaches which deal with many-body systems [Sels and Polkovnikov, PNAS 114 (20) E3909]. Our work shows that the choice of an appropriate ansatz allows to create an efficient STA for interacting systems, which can be effective for short timescales. Due to the approximate nature of the ansatz our STA is not perfect and will, in particular, fail for very fast ramp times. In Fig. 2(a) in the manuscript we show the amount of irreversible work created during the STA and explain that it diverges for times $t_f \leq 1$, as at this point the STA approaches negative values and possesses a more oscillatory form rather than the smooth and slow increase shown in Fig. 1(b). To highlight this fact we have modified Fig. 1(b) to include the STA pulse for $t_f=1$, and include even more examples of STA ramps in the attached Figure R1. Irregardless of the final interaction $g_f$ the STA always gives good results for $t_f>1$, and we have modified the text in the manuscript to emphasize this point.

2) The referee is correct and our approach also works for finite initial g, as good functions for the ansatz are known. Actually, the STA works even better as the initial repulsive interaction between the particles introduces a degree of stiffness to the system which can allow for faster ramps of the interaction compared to starting from $g=0$. The attached Figure R2 shows the irreversible work for a ramp from $g_i=1$ to $g_f=20$, which confirms the improvement over the reference ramp. A sentence has been added to the conclusions of the manuscript to highlight this point.

3) The solutions to the Euler-Lagrange equations allow us to derive interaction ramps as a function of the ramp duration, $t_f$, which means that the shape of the ramp does change for different choices of $t_f$. In response to the first point raised by the referee, we have already modified Fig. 1(b) in the manuscript to highlight this. In general one finds that for larger $t_f$ the change of $g(t)$ is a continuous increase, initially slow and becoming faster towards the end of the ramp. For shorter ramp times the $g(t)$ starts to become oscillatory, initially increasing, going through two turning points and then finally ramping up quickly towards $t_f$ (see $t_f=1$ in Figure R1). For ramping times faster than $t_f=1$ the $g(t)$ does become negative and starts to increase in magnitude (see $t_f=0.5$ in Figure R1). It is on these timescales that the STA starts to fail, as too many dynamical excitations are created and cannot be compensated by our approximate protocol. We have expanded the text in the manuscript that discusses this behaviour. The referee correctly points out that fast protocols that require a lot of changes are difficult to experimentally implement perfectly and we therefore welcome the suggestion to check the robustness of our protocol. In Fig. R3a and R4a we show the interaction ramp for $g_f=5$ and $g_f=20$ respectively. We show the ramps following two paths, one which overshoots (red dashed line) and one which undershoots (blue dotted line) the calculated STA (black solid line). We construct these ramps by adding or subtracting the derivative of the STA (whose maximum is normalised to 1) given by $\tilde{g}(t)=g(t) \pm \gamma \frac{d g(t)}{dt} g_f$, which has the largest deviation when the interaction ramp changes quickly. We scale this error by a factor $\gamma=0.1$ (at most the “error” is a 10% deviation of the final interaction) and calculate the irreversible work (see Fig. R3b and R4b). We see that the irreversible work created does not change significantly, with a slightly larger deviation seen for $g_f=5$, although this is still a less than $10\%$ increase in irreversible work. As long as the interaction ramp approximately follows the path of the STA it will be successful, which highlights the robustness of our technique.

4) In the manuscript we calculate the continuous variable entanglement between the position coordinates of particle 1 ($x_1$) and particle 2 ($x_2$). This is of course different to quantifying the entanglement in the basis of left and right modes of a physically split system, but the continuous variable entanglement of 2 particles is no less valid and is well established [Sun, Zhou, and You, PRA 73, 012336; Wang, Law and Chu, PRA 72, 022346]. By tracing out one of the particles the eigenvalues of the reduced single particle density matrix (RSPDM) then describe the Schmidt rank of the state, that is the degree of mixedness of the reduced state. If the Schmidt rank is greater than one (there are more than one finite eigenvalue of the RSPDM) the state is entangled, and for bipartite pure states the von Neumann entropy is a well defined measure of entanglement, being zero if and only if the state is a product state [Paskauskas and You, PRA 64 042310; Ghirardi and Marinatto, PRA 70, 012109]. The description of the continuous variable entanglement has been expanded upon in the paper.

5) We choose the reference ramp that begins and ends smoothly in an effort to reduce non-equilibrium excitations during the beginning and end of the ramping procedure. The specific form of the reference is not unique and any similarly shaped pulse may be employed. However it shows clear advantages over the linear ramp (see attached figure R5) as it approaches adiabaticity at a faster rate and the irreversible work smoothly decays with increasing ramp time. In comparison, the linear ramp reaches adiabaticity at a much slower rate and can exhibit oscillations in the irreversible work due to resonantly driving the system. Therefore, choosing a reference ramp like the one we consider (or any other smoothly changing function) gives more consistent and better results but also offers a genuinely good comparison to the STA. A line has been added to the manuscript to emphasise this point.

The minor issues raised have also been addressed in the new version of the manuscript.

Attachment:

Figures.pdf

---

## Round 2 · Referee Report · Anonymous (Referee 1) · 2018-12-2

Strengths

Same as in v1.

Weaknesses

Same as in v1.

Report

The authors have taking into account my previous report and I am pleased with their answer.

---

## Round 2 · Author Response

Dear Editor,

We would like to thank the referees for refereeing our manuscript and their detailed reports, and we are grateful that they find our work both original and significant. We have replied to each point raised by the referees in their reports, and have revised our manuscript based on their suggestions. We believe the updated version of the manuscript has improved clarity and we hope that it is accepted for publication in SciPost.

Regards

Thomas Fogarty (On behalf of the authors)

---

## Round 2 · List of Changes

• A sentence has been added to the introduction (Page 2) which emphasizes the approximate nature of our approach.
  • Figure 1 (page 4) has been modified to include colorbars (panel (a) and (c)) and an extra STA ramp has be added to panel (b). The caption has been updated to reflect these changes.
  • The discussion about the choice of reference ramp has been expanded upon (page 5)
  • The discussion about the shape of the STA ramps for different ramp times has been significantly revised (end of page 5 and start of page 6) with an explanation for the modulations of the STA for fast ramp times.
  • The range of validity of the STA has been added (end of page 6 and beginning of page 7)
  • Section 4 "Efficient creation of Entanglement" has been significantly revised and includes a more detailed explanation of the entanglement considered and additional references.
  • Figure 3 (page 8) has been updated to include a colorbar for panels (c) and (d). The caption has been modified to reflect this.
  • The conclusions have been significantly rewritten to include discussions on the validity of the variational approach and the applicability of our protocol for interaction ramps with different initial states. Additional references have been added for the discussion of scaling the shortcut to larger systems and for applications of our approach to collisional quantum gates.

---

## Editorial Decision

published